# *Porphyromonas gingivalis* HmuY and *Streptococcus gordonii* GAPDH—Novel Heme Acquisition Strategy in the Oral Microbiome

**DOI:** 10.3390/ijms21114150

**Published:** 2020-06-10

**Authors:** Paulina Ślęzak, Michał Śmiga, John W. Smalley, Klaudia Siemińska, Teresa Olczak

**Affiliations:** 1Laboratory of Medical Biology, Faculty of Biotechnology, University of Wrocław, 14A F. Joliot-Curie St., 50-383 Wrocław, Poland; paulina.stepien2@uwr.edu.pl (P.Ś.); michal.smiga@uwr.edu.pl (M.Ś.); klaudia.sieminska@uwr.edu.pl (K.S.); 2School of Dentistry, Institute of Clinical Sciences, University of Liverpool, Daulby St., Liverpool L69 3GN, UK; josmall@liv.ac.uk

**Keywords:** *Porphyromonas gingivalis*, *Streptococcus gordonii*, HmuY, GAPDH, microbiome, heme, periodontitis

## Abstract

The oral cavity of healthy individuals is inhabited by commensals, with species of *Streptococcus* being the most abundant and prevalent in sites not affected by periodontal diseases. The development of chronic periodontitis is linked with the environmental shift in the oral microbiome, leading to the domination of periodontopathogens. Structure-function studies showed that *Streptococcus gordonii* employs a “moonlighting” protein glyceraldehyde-3-phosphate dehydrogenase (SgGAPDH) to bind heme, thus forming a heme reservoir for exchange with other proteins. Secreted or surface-associated SgGAPDH coordinates Fe(III)heme using His43. Hemophore-like heme-binding proteins of *Porphyromonas gingivalis* (HmuY), *Prevotella intermedia* (PinO) and *Tannerella forsythia* (Tfo) sequester heme complexed to SgGAPDH. Co-culturing of *P. gingivalis* with *S. gordonii* results in increased *hmuY* gene expression, indicating that HmuY might be required for efficient inter-bacterial interactions. In contrast to the Δ*hmuY* mutant strain, the wild type strain acquires heme and forms deeper biofilm structures on blood agar plates pre-grown with *S. gordonii*. Therefore, our novel paradigm of heme acquisition used by *P. gingivalis* appears to extend to co-infections with other oral bacteria and offers a mechanism for the ability of periodontopathogens to obtain sufficient heme in the host environment. Importantly, *P. gingivalis* is advantaged in terms of acquiring heme, which is vital for its growth survival and virulence.

## 1. Introduction

The oral cavity of healthy individuals is inhabited by commensal microbiota, with species of *Streptococcus* being the most abundant and prevalent in sites not affected by periodontal diseases [1,2]. *Streptococcus gordonii*, a member of the viridans group of streptococci, is one of the initial Gram-positive colonizers of oral mucous surfaces [3]. Despite being a part of the normal microflora, the viridans streptococci (e.g., *S. gordonii* FSS2) are primary etiologic agents of infective endocarditis [4,5]. The development of periodontal diseases is linked with an environmental shift in the oral microbiome, leading to the domination of Gram-negative pathogenic bacteria over early Gram-positive commensal colonizers [6]. In the initial stages of periodontitis, the early Gram-negative anaerobic colonizers (e.g., *Prevotella intermedia*) are responsible for the initiation and progression of periodontitis. Advanced chronic periodontitis is characterized by an increased number of late Gram-negative anaerobic colonizers, mainly *Porphyromonas gingivalis*, *Tannerella forsythia,* and *Treponema denticola* [7]. Importantly, recent studies demonstrated that several oral bacteria are associated with systemic diseases, such as diabetes, rheumatoid arthritis, atherosclerosis, cardiovascular, respiratory diseases, and Alzheimer’s disease [8,9,10].

There is increasing evidence that the majority of bacterial species employ one or more “moonlighting” proteins to aid colonization and induce disease [11,12,13]. Among such proteins is glyceraldehyde-3-phosphate dehydrogenase (GAPDH; EC 1.2.1.12), which converts glyceraldehyde-3-phosphate into glycerate-1,3-biphosphate in the presence of NAD coenzyme [14,15]. The protein is highly conserved across all kingdoms of life, constitutively expressed, present in the cells at high amounts (5–15% of total soluble proteins), and therefore is used as a cytosolic control marker in protein and gene expression studies, and as a reference loading control in biochemical analyses. Besides the housekeeping function in the glycolytic pathway, eukaryotic GAPDH is involved in several other processes, such as transcription regulation, oxidative stress, vesicular trafficking, apoptosis, autophagy [13,14,16]. These functions are regulated, i.e., by protein subunit organization, inter-molecular interactions, variations in cellular localization. Prokaryotic GAPDH is also versatile and some of its functions include binding to mammalian proteins, receptor-mediated acquisition of transferrin- and lactoferrin-bound iron, cell communication [17,18]. Eukaryotic GAPDH shuttles from the cytosol into cellular compartments [19], whereas prokaryotic GAPDH is released upon bacterial lysis or is exported to the bacterial cell wall and the extracellular space, where re-associates itself with the cell wall of viable bacteria [20]. In the secreted or surface-associated form, prokaryotic GAPDH affects the redox balance of the host cell leading to the creation of more favorable environmental conditions for bacterial cells, as well as acting as a binding partner with human extracellular matrix molecules, thus increasing bacterial adherence and host colonization [21,22]. Therefore, an extracellular form of GAPDH, a major secreted protein of *S. gordonii* FSS2, is considered as one of its important virulence factors [4,5,11].

The overall structure of GAPDH consists of identical subunits conserved among different species. The protein exists in the form of homotetramer (~148 kDa), homodimer (~74 kDa), and monomer (~37 kDa) [23], and contains two major domains; the N-terminal NAD binding domain and the C-terminal catalytic or glyceraldehyde-3-phosphate domain [24,25,26,27,28]. Recently, it has been demonstrated that GAPDH belongs to non-canonical heme-binding proteins and binds this cofactor in a reversible process, which often requires the switching of the axial ligands coordinating the heme moiety [29]. Human or rabbit GAPDH binds heme using conserved His53 or His51, respectively, with a heme:tetramer ratio of 1:1, which results in a mono- or bis-coordinate heme complex, ferric heme being bound with higher affinity [19,29]. Therefore, it has been postulated that GAPDH may act as a general transient heme carrier protein in vivo, whereby a redox-based change in heme affinity could facilitate the release of heme from GAPDH in cells [29]. It has been demonstrated that *Streptococcus pneumoniae* secretes GAPDH, which binds both heme and hemoglobin [30], and GAPDH from *Streptococcus pyogenes* is regulated by iron conditions, being tightly associated with the cell surface under iron starvation [31].

It is noteworthy in the context of the characteristics of heme and hemoglobin binding by GAPDH, that *S. gordonii* is able to promote the conversion of oxyhemoglobin into methemoglobin via the action of hydrogen peroxide [32]. When in the methemoglobin form, the affinity for its ferric heme is much reduced [33] and which can more easily be sequestered by serum albumin and by the HmuY hemophore-like heme-binding protein of *P. gingivalis* [34]. Thus, it is likely that GAPDH may play an important role in binding free heme and in the extraction of this cofactor from other hemoproteins. The aim of this study, therefore, was to characterize *S. gordonii* GAPDH (SgGAPDH) in regard to its novel “moonlighting” function related to heme acquisition in the oral microbiome. We hypothesize that the heme-binding ability of SgGAPDH may contribute to the production of a significant pool of heme available for periodontopathogens, thus enabling additional mechanisms to aid their successful colonization, and initiation and progression of chronic periodontitis.

## 2. Results and Discussion

### 2.1. S. gordonii SgGAPDH Binds Heme

A comparison of amino acid sequences of known GAPDH proteins has shown a wide range of identity (46–92%), but crystallographic analysis has revealed almost identical three-dimensional protein structures. Our theoretical analysis of SgGAPDH corroborated those findings (Figure 1, Figure 2 and Appendix A). It has been also demonstrated that eukaryotic GAPDH proteins bind heme in a non-canonical manner, typical for heme-binding proteins involved in heme sensing and transport [29,35,36].

We showed that the Soret maximum determined for SgGAPDH-Fe(III)heme complex was present at 412 nm, and Q band maxima at 536 and 557 nm (Figure 3a). In addition, a charge transfer band at 622 nm and a smaller peak in the Soret region at 351 nm were also visible. The difference spectrum analysis demonstrated four well-resolved peaks at 416, 536, 559, and 653 nm (Figure 3b). After reduction with sodium dithionite, the Soret peak maximum of SgGAPDH red-shifted to 422 nm (Figure 3a). In addition, weak and less resolved Q bands and an additional band at 382 nm could be observed. The difference spectrum analysis showed clear maxima at 425, 529, and 559 nm (Figure 3b). These results support the ability of the SgGAPDH to bind heme, preferentially under oxidizing conditions.

The binding stoichiometry of heme to SgGAPDH was 1:1 (Appendix A), which is similar to that for *S. pneumoniae* GAPDH [30]. Also, the heme dissociation constant value for SgGAPDH (*K*_d_ = 1.7 × 10^−6^ M ± 2.8 × 10^−7^ M) was similar to those reported for well-characterized eukaryotic GAPDH proteins (i.e., from 10^−6^ M to 10^−8^ M) [19,29].

Amino acid comparison of selected GAPDH proteins showed that SgGAPDH possesses conserved histidine residues, except that corresponding to conserved His53 or His51 in human or rabbit GAPDH, respectively (Figure 1, Appendix A). In contrast to eukaryotic GAPDH, SgGAPDH contains the region (KTVVFNTNH) (Figure 1), which is similar to heme-binding motifs exposed on the surface of several non-canonical heme-binding regulatory proteins [30,37]. To study the effects of specific amino acids on heme iron coordination in SgGAPDH, amino acid residues, which could bind heme, were first theoretically identified. Such an approach was based on amino acid sequence comparisons (Figure 1), available crystallographic data of GAPDH proteins, and theoretical modeling of SgGAPDH (Figure 2, Appendix A). Selected conserved His residues were then systematically replaced by an Ala residue, and the ability of the purified protein variants to bind heme was analyzed. The stability and integrity of the purified wild type SgGAPDH and site-directed mutagenesis protein variants were confirmed by SDS-PAGE (Figure 4) and CD spectroscopy (Appendix A). His179 and His192 are located in the NAD-binding pocket (Figure 2), suggesting that they might not be engaged in heme iron coordination. Indeed, as shown in Figure 4, the replacement of these amino acids by Ala did not influence the heme-binding ability. A similar effect was observed in the case of His137 (Figure 4). It is worth noting that the protein variant His109Ala was highly unstable during overexpression and purification, suggesting significant changes in the tertiary protein structure. However, the heme-binding ability was still observed by visual inspection of red-colored protein samples. In addition, His43 and His111 are predicted to be located on the surface of the theoretical SgGAPDH protein structure and may participate in heme binding (Figure 2). As shown in Figure 4, only the SgGAPDH His43Ala protein variant was significantly affected in heme-binding ability, suggesting engagement of this amino acid in Fe(III)heme coordination.

In eukaryotic GAPDH proteins (human and rabbit GAPDH), one heme molecule is bound to the GAPDH tetramer, whereas in prokaryotic GAPDH proteins (GAPDH from *S. pneumoniae*) heme is usually bound to the monomer [29,30,36]. The three-dimensional protein structure solved by crystallographic analysis also suggests tetramer formation in the case of *Streptococcus agalactiae* GAPDH [25,28]. Therefore, we theoretically modeled the SgGAPDH protein structure to examine the possibility of heme coordination through two His residues located in separated monomers. As shown in Figure 3, only His43 might be able to form such binding because His111 is exposed in the SgGAPDH regions that are not involved in potential tetramer formation. However, our experimental analysis demonstrated that SgGAPDH binds one heme molecule to a monomer rather than to a tetramer (Appendix A). Based on our data, one may conclude that monomeric SgGAPDH coordinates heme iron using His43 only.

### 2.2. S. gordonii SgGAPDH Could be a Player in Heme Acquisition Strategy Used by Periodontopathogens

To obtain a desired breakthrough in the fields of understanding and treatment of chronic periodontitis, as well as the roles played by periodontopathogens in systemic diseases, it is necessary to characterize their proteins that are crucial to growth survival and virulence. We have shown that *P. gingivalis* displays a novel paradigm in its heme acquisition from hemoglobin [34], whereby oxyhemoglobin must be firstly oxidized to methemoglobin. In the case of *P. gingivalis*, this process employs the arginine-specific gingipain protease A (HRgpA) [38], and in the case of *P. intermedia* the protease interpain A (InpA) [39]. The bacteria are then able to fully proteolyze the more susceptible methemoglobin substrate to release free heme, mostly through *P. gingivalis* lysine-specific gingipain K (Kgp) activity and via heme sequestration mediated by the HmuY protein [40,41,42]. The latter protein is the first representative of a novel family of hemophore-like proteins which have been recently characterized and which include Tfo produced by *T. forsythia*, as well as PinO and PinA produced by *P. intermedia* [34,40,42,43]. These proteins are similar to classical hemophores used by *Serratia marcescens* [44] or *Yersinia pestis* [45], which bind heme with high affinity and deliver it to the outer-membrane receptors for subsequent transport into the cell. However, in contrast to secreted hemophores, HmuY and its homologs are associated with the outer membrane and can be shed from the bacterial cell surface by limited proteolytic processing [40,42,43].

Our novel paradigm of heme acquisition, which is displayed by the black-pigmented anaerobes, appears to extend also to co-infections with other bacteria and offers a synergistic mechanism for the ability of *P. gingivalis* to obtain sufficient heme in the host environment. We showed that the presence of *Pseudomonas aeruginosa* pyocyanin facilitates the extraction of heme from hemoglobin by the *P. gingivalis* HmuY by oxidizing oxyhemoglobin to methemoglobin [46]. Moreover, we demonstrated that HmuY homologs produced by *T. forsythia* (Tfo) and *P. intermedia* (PinO and PinA) also bind heme, but importantly the heme-bound proteins may provide the *P. gingivalis* HmuY with heme, resulting in the increased virulence of the latter species [42,43].

It has been demonstrated that *S. gordonii* produces hydrogen peroxide, which converts oxyhemoglobin into methemoglobin, both under microaerobic and anaerobic conditions [32,47]. Therefore, the ability of *S. gordonii* to mediate hemoglobin oxidation may support heme acquisition during co-aggregation with *P. gingivalis* [32]. *S. gordonii* may increase heme availability by promoting hemolysis and formation of methemoglobin, which can be utilized as a heme source via proteolytic degradation by the action of *P. gingivalis* Kgp or via extraction by the HmuY [34]. Indeed, we have previously shown that HmuY is able to sequester heme bound to methemoglobin, the latter formed by hydrogen peroxide produced by *S. gordonii* [32]. Importantly, the HmuY-heme complex formed was stable at the highest concentrations of hydrogen peroxide produced by this bacterium [32]. In this study, we showed that SgGAPDH is not able to sequester heme bound to methemoglobin (Appendix A) and may, at least in part, exchange the heme bound to serum albumin (Appendix A). It is possible that in the early phase of disease in the presence of abundant gingival crevicular fluid which contains albumin as its major serum protein [40], any free heme may remain bound to albumin, and any heme pre-bound to the SgGAPDH could be transferred to albumin, thus forming a dynamic heme reservoir, from which the heme might be easily accessed by periodontopathogens as a result of bacterial protease activity or hemophore-like activity of HmuY and its homologs [38,39,42,43].

To assess any possible syntrophy between *P. gingivalis* HmuY or its homologs and *S. gordonii* SgGAPDH, we first examined the interactions between HmuY, PinO, PinA, Tfo apo-proteins, and SgGAPDH-heme complex. As shown in Figure 5, HmuY, PinO and Tfo sequestered Fe(III)heme which had been complexed to SgGAPDH. In contrast, *P. intermedia* PinA was not able to sequester heme bound to SgGAPDH. In the case of Fe(II)heme, all the hemophore-like proteins examined were able to sequester heme bound to SgGAPDH (Figure 6). Under air (oxidizing) conditions apo-SgGAPDH was able to capture heme bound only to PinA (Appendix A). It is worth noting, however, that if *P. intermedia* grows more abundantly in the early phase of disease it could degrade both the albumin-heme complex and the SgGAPDH-heme complex (Figure 6). Under reducing conditions, apo-SgGAPDH was not able to sequester heme bound to any of the hemophore-like proteins examined (Appendix A). Therefore, we hypothesize that heme bound to secreted or surface-associated SgGAPDH might represent a heme reservoir for *P. gingivalis* or other periodontopathogens.

Our results confirmed that SgGAPDH is exposed on the cell surface of *S. gordonii* as well as is being secreted into the culture medium (Figure 7). Therefore, the heme bound to SgGAPDH might be accessed by *P. intermedia* in the early phases of colonization when *S. gordonii* and *P. intermedia* dominate over late colonizers, but that later, as more anaerobic and reducing conditions develop in the periodontal pocket, it could be used by HmuY and Tfo, aiding colonization of both *P. gingivalis* and *T. forsythia*. It is worth noting that viridans streptococci secrete higher levels of GAPDH when the pH increases [48], which could be advantageous for periodontopathogens, especially for the late colonizers residing in deep periodontal pockets where the environmental pH is increased above neutrality [49]. Experiments demonstrating the growth of selected bacteria in the presence of the purified proteins showed that although SgGAPDH was not proteolytically digested during growth in *S. gordonii* and *T. forsythia* cultures, proteases produced by *P. gingivalis* and *P. intermedia* caused its degradation (Appendix A), additionally aiding heme release. Our data suggest that periodontitis-associated bacteria, especially the better adjusted to both anaerobic and aerobic conditions *P. gingivalis*, could also benefit from commensal members of the oral microbiome. This hypothesis is supported, at least in part, by the observation of co-localized *P. gingivalis* and *S. gordonii* in various layers of supragingival plaque [50].

### 2.3. P. gingivalis HmuY May Participate in Co-Aggregation and Biofilm Formation with S. gordonii

Streptococci occupy a broad range of oral habitats where they co-localize and co-aggregate with other bacteria [50,51,52]. There is growing evidence demonstrating that interaction between *P. gingivalis* and oral streptococci is crucial for biofilm formation and colonization of *P. gingivalis* [50,53,54,55,56,57,58]. Interaction between *S. gordonii* and *P. gingivalis* employs the major (FimA) and minor fimbrial (Mfa1) proteins of *P. gingivalis* and outer-membrane adhesins (SspA and SspB) of *S. gordonii* [58,59,60,61,62,63,64,65,66]. In addition, in vitro studies showed that *S. gordonii* might promote the penetration and invasion of dentinal tubules by *P. gingivalis* by providing an adhesive platform on intra-tubular dentine collagen [67]. It has been suggested, however, that those interactions may not be sufficient for the formation of biofilms composed of *S. gordonii* and *P. gingivalis*. In this study, we corroborated the co-aggregation ability occurring between *S. gordonii* and selected periodontopathogens (Figure 8). Interestingly, we showed that such inter-bacterial interaction might employ the HmuY protein. As shown in Figure 9, the Δ*hmuY* mutant TO4 strain exhibited a lower co-aggregation tendency as compared with the wild type A7436 strain. Recently, we demonstrated that HmuY homologs produced by *P. intermedia* and *T. forsythia* bind heme preferentially under anaerobic conditions, suggesting their employment in heme acquisition in deep periodontal pockets [42,43]. In contrast to Tfo, PinO, and PinA, HmuY is very efficient in heme binding under both reducing and air (oxidizing) conditions [41,42]. Although *P. gingivalis* is an anaerobe, the bacterium can also grow in microbiome structures exposed to aerobic conditions [50]. Therefore, co-localization of *P. gingivalis* and *S. gordonii* could be beneficial for *P. gingivalis* growing in the oral microbiome under a range of higher oxygen levels in which it may not otherwise survive alone.

Visual inspection demonstrated that, in contrast to the Δ*hmuY* mutant strain, the wild type *P. gingivalis* strain grew faster and formed bigger biofilm structures on blood agar plates pre-grown with *S. gordonii* (Figure 10). Interestingly, we also observed a different ability to form pigment. The Δ*hmuY* mutant strain became black pigmented faster, whereas the wild type strain, although forming pigment, did so but with significantly lesser ability, even after prolonged growth (Figure 10). We suspect that such a feature could result from higher expression and/or better exposure of hemagglutinin domains present in proteins, which are also known to participate in pigment formation [34].

Co-culture of *P. gingivalis* with *S. gordonii* resulted in increased *hmuY* gene expression (fold change of 3.2 ± 0.45 as determined by RT-qPCR), indicating that HmuY might be required for efficient inter-bacterial interactions in heme uptake by *P. gingivalis*. Other studies also demonstrated increased expression of the *hmuY* gene during synergistic interactions between bacterial members in the dental plaque of periodontitis patients [68]. In contrast, decreased expression of the *hmuY* gene was found by others testing *P. gingivalis* and *S. gordonii* co-cultures, but in those studies the less virulent *P. gingivalis* strain ATCC 33277 and different culturing conditions were used [69]. A decreased co-aggregation ability was also observed in this study in the case of Δ*hmuR* mutant WS1 strain and the double Δ*hmuY*Δ*hmuR* mutant TO2 strains (Figure 9). Although the HmuR protein was suggested to be important for efficient community formation between *P. gingivalis* and *Fusobacterium nucleatum* [64], no significant change in the *hmuR* gene expression was observed in our study (fold change of −1.22 ± 0.51 as determined by RT-qPCR) during co-culture of *P. gingivalis* and *S. gordonii*. Interestingly, when the Δ*hmuR* (WS1) and Δ*hmuY*Δ*hmuR* (TO2) mutant strains were subjected to growth on blood agar plates pre-grown with *S. gordonii*, slower biofilm formation but significantly faster pigmentation was observed as compared to the wild type (A7436) or Δ*hmuY* mutant (TO4) strain (Figure 10). One explanation for this could be the lack of the outer-membrane HmuR transporter, which causes accumulation of the pigment on the cell surface instead of heme uptake. Our data may suggest that the *hmu* operon, with a leading role played by the *hmuY* gene, may be important for efficient host colonization by *P. gingivalis*.

## 3. Materials and Methods

### 3.1. Bacterial Strains and Growth Conditions

*P. gingivalis* wild type (A7436), Δ*hmuY* (TO4), Δ*hmuR* (WS1), Δ*hmuY*Δ*hmuR* (TO2) mutant strains, *P. intermedia* 17, and *T. forsythia* ATCC 43037 were grown anaerobically at 37 °C for 5–10 days on blood agar plates composed of Schaedler broth (containing hemin and L-cysteine), and supplemented with 5% sheep blood and menadione (Biomaxima, Lublin, Poland). These cultures were inoculated into liquid basal medium (BM) prepared of 3% trypticase soy broth (Becton Dickinson, Sparks, MD, USA), 0.5% yeast extract (Biomaxima), 0.5 mg/L menadione (Fluka, Munich, Germany), and 0.05% L-cysteine (Sigma, St. Louis, MO, USA). To grow bacteria under high-iron/heme conditions (Hm), BM medium was supplemented with 7.7 µM hemin (ICN, Biomedicals, Aurora, OH, USA), and to grow bacteria under low-iron/heme conditions (DIP), hemin was not added, and iron was chelated by addition of 160 µM 2,2-dipyridyl. *T. forsythia* was cultured in the presence of 10 µg/mL N-acetylmuramic acid (Sigma). *P. gingivalis* mutant TO4 and mutant WS1 strains were cultured in the presence of 1 µg/mL erythromycin, and mutant TO2 strain in the presence of 1 µg/mL erythromycin and 1 µg/mL tetracycline [70,71].

*S. gordonii* ATCC 10558 was grown on blood agar plates or in 5% trypticase soy broth (TSB; Becton Dickinson) under an increased concentration of CO_2_ using the Atmosphere generation system (Thermo Scientific, Rockford, IL, USA) as reported previously [72]. *Escherichia coli* ER2566 (New England Biolabs, Ipswich, MA, USA) and BL21-CodonPlus(DE3)-RIL (Agilent Technologies, Santa Clara, CA, USA) strains were cultured in terrific broth (BioShop, Burlington, ON, Canada) under aerobic conditions.

To examine biofilm formation, *S. gordonii* was grown on blood agar plates for 24 h under an increased concentration of CO_2_. Then, 50 µL of *P. gingivalis* cultures adjusted in PBS to OD_600_ = 1.0 was applied (starting from 5 × 10^7^ cells/50 µL/spot and further diluted) on blood agar plates alone or on blood agar plates pre-grown with *S. gordonii*. Bacterial growth was continued under anaerobic conditions for 12 days. The formation of biofilm structure and pigmentation efficiency were monitored by visual inspection.

### 3.2. Co-Aggregation Assay

Co-aggregation assay was performed as described previously [72]. Briefly, overnight liquid cultures of *P. gingivalis*, *P. intermedia*, *T. forsythia* and *S. gordonii* were centrifuged (4000× *g*, 20 min, 4 °C) and washed with 20 mM sodium phosphate buffer, pH 7.4, containing 140 mM NaCl (PBS). Subsequently, optical density at 600 nm (OD_600_) of the cultures was adjusted to 1.0 using PBS. The cultures of the respective bacteria were mixed at a 1:1 ratio and incubated anaerobically for 6 h. The co-aggregation rate was monitored by measurement of the decrease in OD_600_. The sum of the auto-aggregation of the tested mono-species cultures was used as a control.

### 3.3. Overexpression, Purification and Site-Directed Mutagenesis

The DNA sequence encoding SgGAPDH protein (NCBI accession no. SQF26726.1) was cloned into the NdeI and EcoRV restriction sites of pTXB1 plasmid (New England Biolabs) or into the XcmI and BamHI restriction sites of pTriEx-4 plasmid (Merck, Darmstadt, Germany) using primers listed in Appendix A and NEBuilder^®^HiFi DNA Assembly (New England Biolabs). Resulting pTXB1_SgGAPDH or pTriEx_SgGAPDH expression plasmids were used to overexpress SgGAPDH protein. Recombinant protein possessing C-terminal intein with chitin-binding domain was purified using affinity chromatography with chitin resin (New England Biolabs) according to the manufacturer’s instructions. SgGAPDH protein was released from the resin by on-column cleavage in the presence of 50 mM β-mercaptoethanol at 4 °C for 24 h, followed by affinity chromatography using Cibacron Blue 3G-A (Sigma). The protein was bound to the chromatographic resin in 50 mM Tris/HCl buffer, pH 7.6, containing 30 mM NaCl and eluted using 50 mM Tris/HCl buffer, pH 7.6, containing 1 M NaCl. Recombinant protein possessing N-terminal His-tag with the Factor Xa recognition site was purified using TALON resin (Clontech, Mountain View, CA, USA) according to the manufacturer’s instructions.

Selected amino acids of SgGAPDH with a potential ability to coordinate heme iron were substituted by an alanine, resulting in overexpressing of single point mutation protein variants. Point mutations were introduced into expression plasmids using a QuikChange II XL Site-Directed Mutagenesis Kit (Agilent Technologies) and primers listed in Appendix A. The resulting plasmids were sequenced and used to transform *E. coli* cells.

The *P. gingivalis* A7436 HmuY protein (NCBI accession no. CAM 31898) lacking the signal peptide (MKKIIFSALCALPLIVSLTSC) and additional (GKKK) N-terminal amino acid residues present in the full-length protein sequence was overexpressed using *E. coli* ER2566 cells (New England Biolabs), and purified from a soluble fraction of the *E. coli* cell lysate as described previously [73].

*P. intermedia* 17 PinO (NCBI accession no. AFJ07542) and PinA (NCBI accession no. AFJ08449) proteins, lacking the predicted signal peptides (MKTKIFAVACLATLLFTSC and MKFKSFMALSCLTVLLFSSC, respectively), were overexpressed using a pMALc5x_PinO or a pMALc5x_PinA plasmid and *E. coli* BL21-CodonPlus(DE3)-RIL cells (Agilent Technologies) and purified from a soluble fraction of the *E. coli* cell lysate as described previously [43].

The DNA sequence encoding *T. forsythia* ATCC 43,037 Tfo protein (NCBI accession no. CEH11291), lacking the predicted signal peptide (MKMRNVMTLALVALSLAFVGC) was amplified using primers listed in Appendix A. Then, using NEBuilder^®^ HiFi DNA Assembly (New England Biolabs), the gene was cloned into the XmnI and BamHI restriction sites of pMAL-c5x plasmid, modified as reported previously [74]. Overexpression and purification of Tfo protein were performed as described for PinA and PinO proteins.

The concentration of SgGAPDH protein was determined spectrophotometrically using the empirical molar absorption coefficient (ε_280_ = 23.97 mM^−1^cm^−1^), calculated in this study as reported previously [40].

### 3.4. Susceptibility of SgGAPDH to Proteolytic Digestion

To examine the susceptibility of the SgGAPDH to proteases produced by the examined bacterial species, bacterial cells were grown under rich, high-iron/heme conditions (Hm), ensuring proper cell viability and efficient proteolytic activity [71,73], in the presence of added purified SgGAPDH at final 5 µM concentration [42,43]. All cultures (10 ml) were started at OD_600_ = 0.2 (time 0), grown and collected at 24 h. The number of bacterial cells at the starting point was ~2 × 10^8^ per ml of the culture medium and increased during cultivation. As controls, culture media with the addition of the purified protein were analyzed. At the indicated time points, aliquots of samples were examined by SDS-PAGE and Coomassie Brilliant Blue G-250 staining [73].

### 3.5. Analysis of Heme Binding

Heme (hemin chloride; ICN Biomedicals) solutions were prepared as reported previously [38,42]. The formation of the heme-protein complexes was examined in PBS or 100 mM Tris–HCl buffer, pH 7.5, containing 140 mM NaCl (TBS). UV-visible spectra were recorded in the range 250–700 nm with a double beam Jasco V-650 spectrophotometer using cuvettes with 10 mm path length. Titration curves were analyzed using the equation for a one-site binding model, and dissociation constant (*K*_d_) values determined as reported earlier [42] using OriginPro 8 software (OriginLab, Northampton, MA, USA).

To determine heme binding ability of site-directed mutagenesis variants of SgGAPDH, 5 µM protein was mixed with 4 µM Fe(III)heme, and spectra were recorded and compared with spectra obtained for the wild type protein.

To confirm the heme to protein binding ratio, an additional approach was applied. For this purpose, SgGAPDH was saturated with Fe(III)-heme and the excess heme was removed by desalting on a Zeba Spin Desalting Column 7 K MWCO (Thermo Scientific). 4 µM SgGAPDH-Fe(III)heme complex was mixed with 25 µM apo-HmuY in PBS, incubated for 30 minutes, and the UV-visible spectrum was recorded. In parallel, 25 µM apo-HmuY in the presence of 4 µM apo-SgGAPDH was titrated with Fe(III)heme. The UV-visible spectrum was examined after increasing the heme concentration in the sample by 1 µM. The absorbance at 411 nm that corresponds to the HmuY-Fe(III)heme complex Soret maximum was used to create a standard curve of the amount of heme bound to the HmuY protein. Excess HmuY protein in the sample ensured that all heme would be sequestered from SgGAPDH. The amount of heme bound to the SgGAPDH monomer was determined using the standard curve.

To analyze the redox properties of the heme with iron in the Fe(II) state, 10 mM sodium dithionite was used as the reductant with or without mineral oil overlay [38,40].

### 3.6. Detection of SgGAPDH Protein

To detect SgGAPDH present on the *S. gordonii* cell surface or secreted into the culture medium, enzyme-linked immunosorbent assay (ELISA) was first used. Bacteria were grown in 5% TSB under an increased concentration of CO_2_ for 24 h, centrifuged (4000× *g*, 12 min), washed with PBS, and suspended in 50 mM Tris/HCl buffer, pH 8.0, to an OD_600_ = 1. Bacteria were examined in the native form or after treatment for 1 h at 37 °C with proteinase K (20 µg/mL) and/or 0.5% SDS. Subsequently, bacteria were washed 3 times with PBS and suspended in PBS to the original volume. The culture medium was filtrated through 0.22 µm filters, ultracentrifuged (100,000× *g*, 2 h), and concentrated ~20× using 10 kDa cut-off filter units (Millipore, Billerica, MA, USA). Then, 96-well polystyrene plates (Sarstedt, Waltham, MA, USA) were coated with either 100 µL of bacterial suspension or 100 µL of concentrated culture medium. The protein was probed with rabbit polyclonal anti-SgGAPDH antibodies, raised in this study against purified SgGAPDH protein (ProteoGenix, Schiltigheim, France), and complexes formed detected using HRP-conjugated anti-rabbit IgG antibodies (Sigma) and a substrate solution as described previously [75].

To confirm the presence of SgGAPDH, Western blotting was employed. Bacterial suspension in the native form and/or treated as described above was subjected to lysozyme digestion (2 µg/mL, 10 min, 37 °C), and sonication (20 sec) in SDS-PAGE denaturation buffer. Proteins present in bacterial cell lysates (20 µL of the OD_600_ = 0.02 samples) or in 40 µL of concentrated culture medium were separated by SDS-PAGE, transferred onto a nitrocellulose membrane (Millipore), probed with rabbit anti-SgGAPDH antibodies, and complexes formed detected using HRP-conjugated anti-rabbit IgG antibodies and chemiluminescence staining (Perkin Elmer, Waltham, MA, USA) [73].

### 3.7. Heme Sequestration Experiments

Co-incubation of HmuY, Tfo, PinO, or PinA with SgGAPDH, each protein in apo- or holo-form, was carried out under air (oxidizing) or reducing (10 mM sodium dithionite) conditions in PBS at 37 °C and monitored by UV-visible spectroscopy using each protein at 10 µM concentration [40,42,43]. Co-incubation of methemoglobin or human serum albumin with SgGAPDH was carried out using each protein at 5 µM concentration. Protein:heme complexes were formed at a 1:1.2 ratio, and the excess heme was removed by desalting with a Zeba Spin Desalting Column 7 K MWCO (Thermo Scientific).

### 3.8. Reverse Transcriptase-Quantitative Polymerase Chain Reaction (RT-qPCR)

Total RNA was purified from 1 ml of *P. gingivalis* culture or 1 ml of *P. gingivalis* and *S. gordonii* co-cultures using the Total RNA Mini Kit (A&A Biotechnology, Gdynia, Poland) and the Clean-Up RNA concentrator kit (A&A Biotechnology). Reverse transcription was carried out using 1 µg of RNA and SensiFAST cDNA synthesis kit (Bioline, London, UK). qPCR was performed using SensiFAST SYBR No-ROX kit (Bioline) and the LightCycler 96 system (Roche, Basel, Switzerland). The amplification reaction was carried out as follows: an initial denaturation at 95 °C for 2 min, 40 cycles of denaturation at 95 °C for 5 s, primer annealing at 60 °C for 10 s, and extension at 72 °C for 20 s. The melting curves were analyzed to monitor the quality of PCR products. Relative quantification of the *hmuY* or *hmuR* gene was determined in comparison to the *16SrRNA* gene of *P. gingivalis* A7436 (PGA7_00000960) as a reference, using the ΔΔC_t_ method and LightCycler 96 software (Roche). All samples and controls were run in triplicate in three independent experiments for the target and reference genes. All primers are listed in Appendix A.

### 3.9. Circular Dichroism (CD) Analysis

Far-UV CD spectroscopy was carried out using a Jasco J-1500 spectropolarimeter (Jasco, Tokyo, Japan). Protein samples at 2.5 μM concentration in 10 mM sodium phosphate buffer, pH 7.5, containing 20 mM NaCl, were examined in a 2-mm path-length cuvette. The spectra were recorded over a wavelength range of 200–260 nm by signal averaging of three spectra. Scanning speed of 100 nm/min, bandwidth 2 nm, and response time 2 s were used. All spectra were baseline corrected for the buffer.

### 3.10. Bioinformatics and Statistical Analyses

The sequences were aligned by the Clustal Omega tool from the EMBL-EBI server (the European Bioinformatics Institute). The statistical analysis was performed using Student’s t or one-way ANOVA test. Data were expressed as mean ± standard deviation (mean ± SD). For statistical analysis, the GraphPad Prism 8.0 software (GraphPad Sooftware Inc., San Diego, CA, USA) was used. The theoretical three-dimensional model of SgGAPDH protein structure was constructed using Phyre2 software [76] based on GAPDH protein structure from *Streptococcus agalactiae* (PDB: 4QX6). Subsequently, the SgGAPDH model was submitted to ModRefiner server [77]. Final protein structure of SgGAPDH protein was visualized by Swiss-PdbViewer [78].

## 4. Conclusions

We are aware that bacteria grown in liquid cultures or solid media under laboratory conditions may display different properties compared with those grown in polymicrobial consortia within dental plaque biofilms. However, the model presented here has allowed for important observations that have demonstrated interactions between *P. gingivalis* and *S. gordonii*. Our observations clearly showed that *P. gingivalis* is advantaged during biofilm co-culture in terms of acquiring heme, which could be vital for its growth survival and virulence.

## Figures and Tables

**Figure 1 ijms-21-04150-f001:**
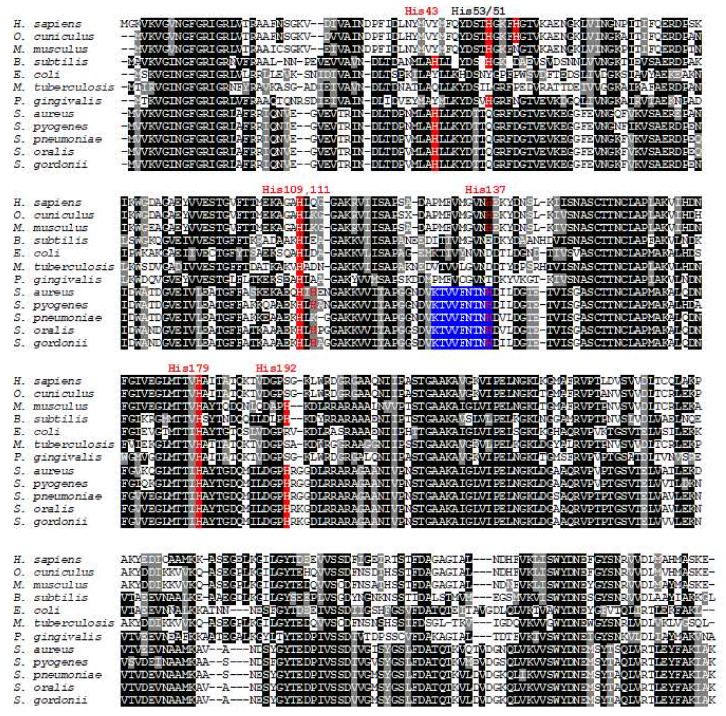
A comparison of amino acid sequences of selected GAPDH proteins. Amino acid residues involved in heme binding in several GAPDH proteins are marked in red. Conserved heme regulatory motif is marked in blue. His53/51 - conserved histidine residue engaged in heme coordination in human and rabbit GAPDH, respectively. His43, His109, His111, His137, His179, and His192 histidine residues replaced singly by an alanine residue in this study in SgGAPDH using site-directed mutagenesis are marked in red. Amino acid sequences were identified using BLAST search and compared using the Clustal Omega and the BoxShade servers.

**Figure 2 ijms-21-04150-f002:**
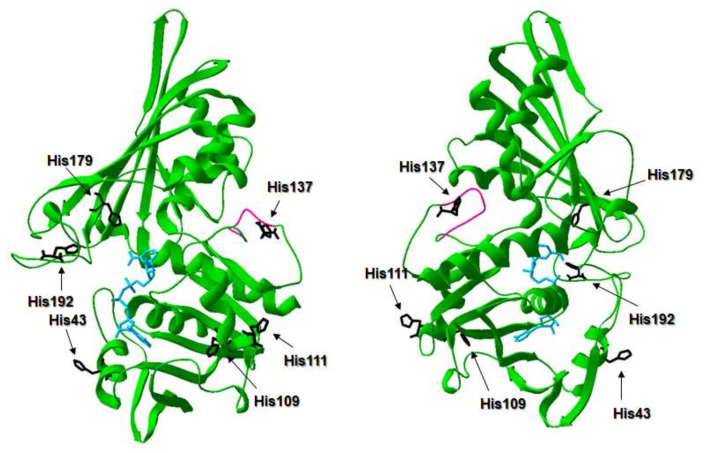
Localization of conserved amino acid residues in the theoretical three-dimensional model of the SgGAPDH protein structure. Potential heme-coordinating histidine residues are shown in black and indicated with arrows, the potential heme-binding regulatory motif is shown in pink, and the NAD molecule is shown in blue. Two orientations of the modeled protein structure are shown.

**Figure 3 ijms-21-04150-f003:**
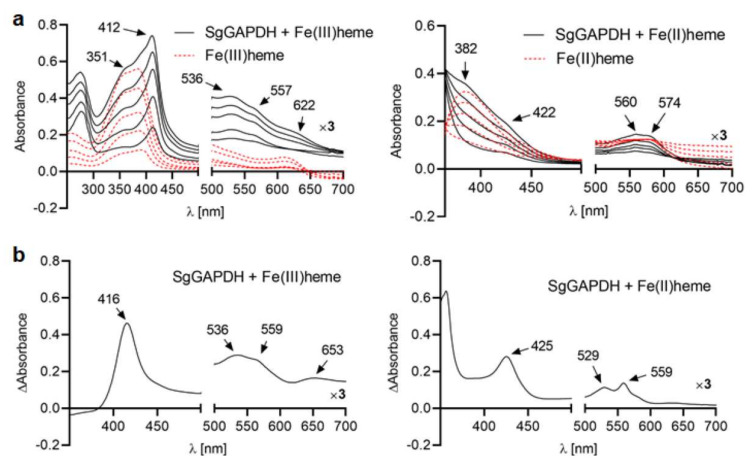
Heme titration of SgGAPDH. (**a**) UV-visible absorption and (**b**) difference spectra recorded after titration of SgGAPDH (10 µM) with heme (final heme concentration in samples: 0–10 µM) are shown. Various lines in (**a**) represent increasing concentrations of heme added to the buffer alone (red dashed lines) or to protein samples (solid black lines). Samples were examined under air (oxidizing) conditions (left panel) or reduced by sodium dithionite (right panel).

**Figure 4 ijms-21-04150-f004:**
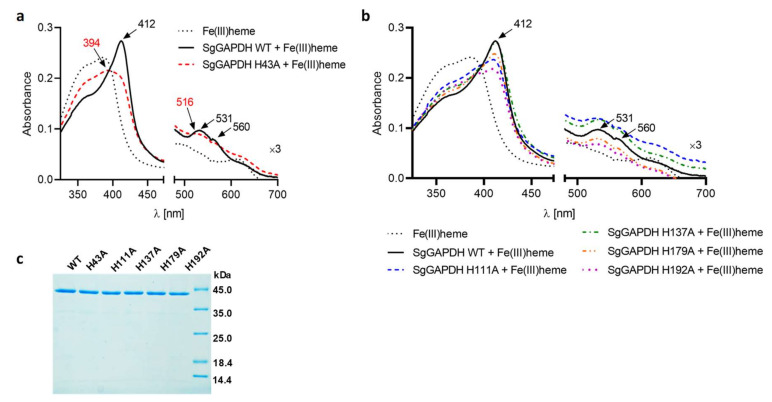
Heme binding to SgGAPDH site-directed mutagenesis protein variants. UV-visible spectra recorded for 5 µM (**a**) His43Ala or (**b**) His111Ala, His137Ala, His179Ala, and His192Ala protein variants in complex with 4 µM Fe(III)heme are shown. (**c**) Confirmation of equal protein amount in all samples was examined by SDS-PAGE and CBB G-250 staining.

**Figure 5 ijms-21-04150-f005:**
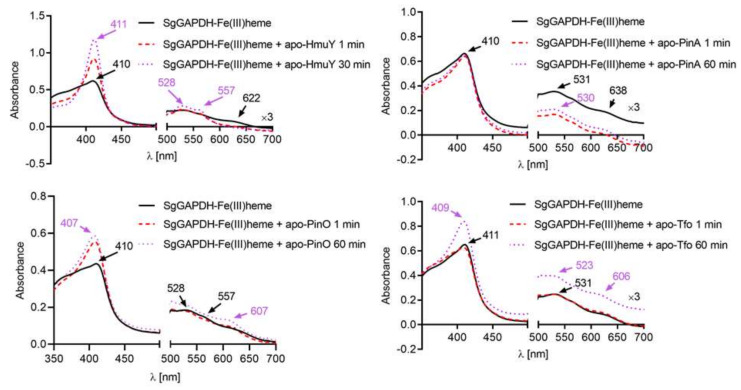
Heme sequestration by HmuY and its homologs from SgGAPDH-heme complex under oxidizing conditions. SgGAPDH-heme complex (10 µM) was incubated under air (oxidizing) conditions with equimolar concentration of HmuY, PinO, PinA or Tfo apo-proteins. Changes in absorption spectra analyzed by UV-visible spectroscopy are shown at indicated time points.

**Figure 6 ijms-21-04150-f006:**
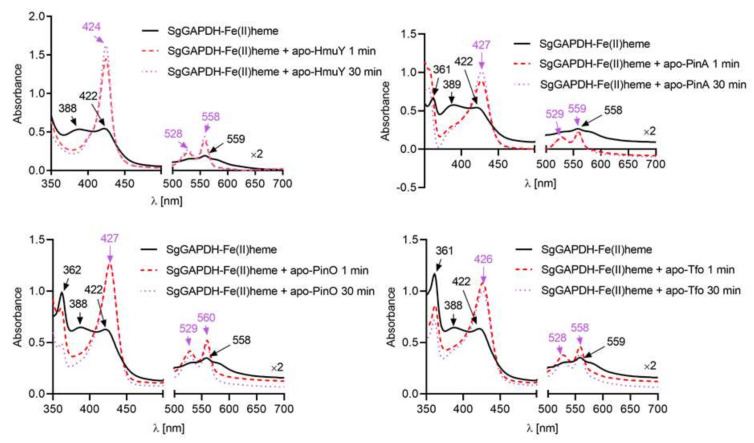
Heme sequestration by HmuY and its homologs from SgGAPDH-heme complex under reducing conditions. SgGAPDH-heme complex (10 µM) was incubated under reducing conditions with an equimolar concentration of HmuY, PinO, PinA, or Tfo apo-proteins. Changes in absorption spectra analyzed by UV-visible spectroscopy are shown at indicated time points.

**Figure 7 ijms-21-04150-f007:**
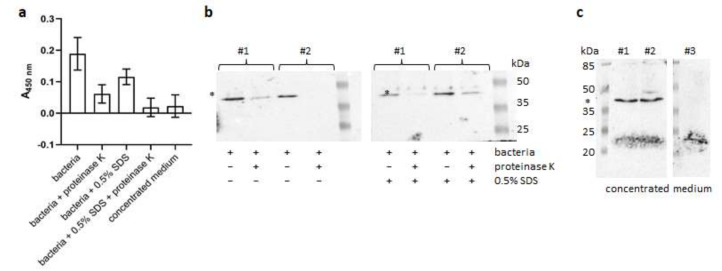
Identification of SgGAPDH forms produced by *S. gordonii*. SgGAPDH was visualized using (**a**) ELISA or (**b**,**c**) Western blotting. The protein was detected on the *S. gordonii* cell surface (**a**,**b**) or in the culture medium (**a**,**c**) using rabbit polyclonal anti-SgGAPDH antibodies and HRP-conjugated anti-rabbit IgG antibodies. The protein associated with the bacterial cell surface was examined in the native form or after treatment with proteinase K and/or SDS. Soluble secreted protein was detected in ~20× concentrated culture medium. (**b**,**c**) SgGAPDH was examined in two independent samples (#1 and #2). (**c**) To examine putative cross-reactivity of anti-SgGAPDH antibodies, 20× concentrated culture medium alone was also examined (sample #3). SgGAPDH protein is indicated by asterisks (∗).

**Figure 8 ijms-21-04150-f008:**
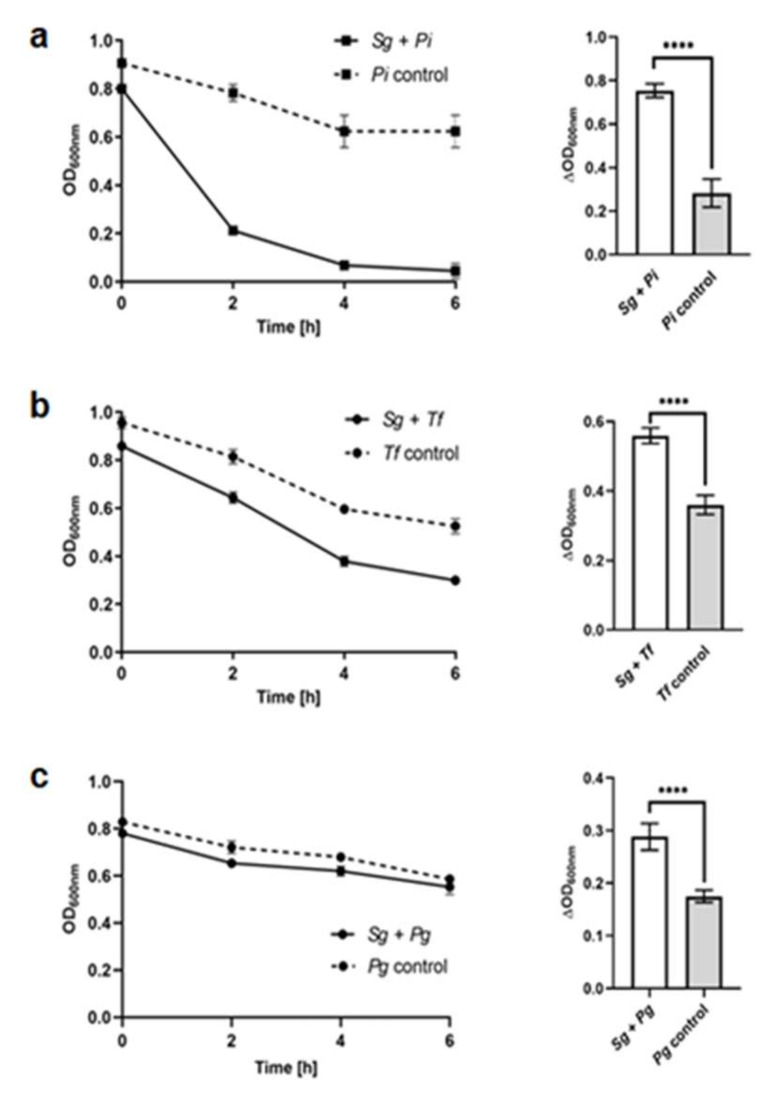
Interaction of *S. gordonii* with periodontopathogens. The tendency of *S. gordonii* to co-aggregate with (**a**) *P. intermedia*, (**b**) *T. forsythia*, and (**c**) *P. gingivalis* was monitored by measuring optical density at 600 nm at the indicated time points (left panel) or after 6 h (right panel). *Pi*, *Prevotella intermedia*; *Tf*, *Tannerella forsythia*; *Pg*, *Porphyromonas gingivalis*; *Sg*, *Streptococcus gordonii*. **** *p* < 0.0001.

**Figure 9 ijms-21-04150-f009:**
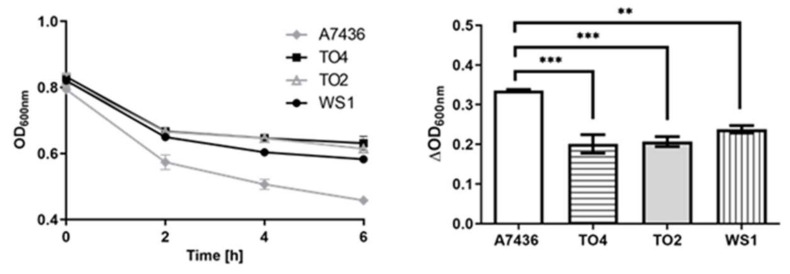
Interaction of *S. gordonii* with *P. gingivalis* strains. The tendency of *S. gordonii* to co-aggregate with *P. gingivalis* wild type (A7436) and mutant strains lacking *hmuY* (TO4), *hmuR* (WS1) and both *hmuY* and *hmuR* genes was monitored by measuring optical density at 600 nm at the indicated time points (left panel) or after 6 h (right panel). A7436, wild type *P. gingivalis* strain; TO4, Δ*hmuY* mutant strain; TO2, Δ*hmuY*Δ*hmuR* mutant strain; WS1, Δ*hmuR* mutant strain. All mutant strains were constructed in the A7436 wild type strain. ** *p* < 0.01; *** *p* < 0.001.

**Figure 10 ijms-21-04150-f010:**
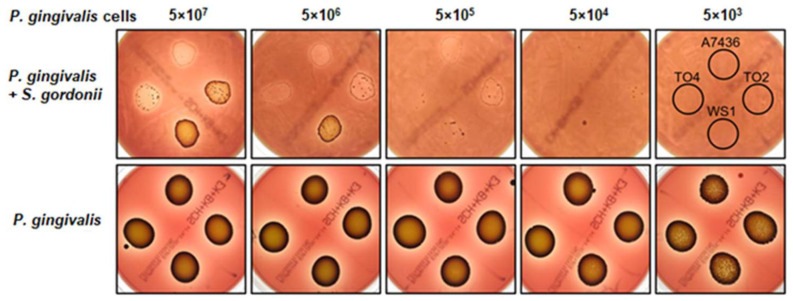
Two-species biofilm formation. *S. gordonii* was grown on blood agar plates for 24 h under an increased concentration of CO_2_. Then, the indicated number of *P. gingivalis* cells was applied on and the growth of *P. gingivalis* wild type strain (A7436) or Δ*hmuY* (TO4), Δ*hmuR* (WS1), Δ*hmuY*Δ*hmuR* (TO2) mutant strains alone (bottom panel) or as co-cultures with *S. gordonii* (top panel) was examined under anaerobic conditions. Biofilm formation and pigmentation were visually inspected.

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
