# Peer review of "Porphyromonas gingivalis* HmuY and *Streptococcus gordonii* GAPDH—Novel Heme Acquisition Strategy in the Oral Microbiome"

_ijms, 2020, doi:10.3390/ijms21114150_

Round 1

Reviewer 1 Report

Dear editor,

Thank you for giving an opportunity to review manuscript “Porphyromonas gingivalis HmuY and Streptococcus gordonii GAPDH – novel heme acquisition strategy in the oral microbiome”. This is well written manuscript clearly showing the novel moonlighting phenomena of glyceraldehyde-3-phosphate dehydrogenase (SgGAPDH) which piles up the heme and exchange with other proteins. Manuscript has convincing data. Followings are my few comments:

  1. Fig. 4 C, these mutant proteins need to be characterized for their stabilities and proper folding before comparing for Heme binding. Common protein characterizations assays like: Differential scanning fluorimetry (DSF) and thermal melting comparison with wild type Versus mutant proteins may give tentative idea.
  2. Fig. 6abc will be strong data sets if SgGAPDH mutant and complemented strains will be included.
  3. Figure 8abc needs sg alone control to see the aggregation pattern without pg, pi and tf.
  4. Figure 10 also needs Sg alone control. This data set also  needs quantitative measurement for biofilm formation and pigmentation.

Reviewer 2 Report

It is known that between oral bacteria are nutrient, substrate and ecological relationships. Authors very good presented one of the relationships between Porphyromonas gingivalis and Streptococcus gordonii. P. gingivalis acquired heme which is significant for its growth survival and virulence. Heme acquisition by P. gingivalis may be from other bacterial species. Authors in their work, confirmed this important bacterial co-operation.

I suggest enlarge figures, because are unclear. In description of figure 2 is word "ale" - for correction.
